# Work Ability and Well-Being Management and Its Barriers and Facilitators in Multinational Organizations: A Scoping Review

**DOI:** 10.3390/healthcare11070978

**Published:** 2023-03-29

**Authors:** Rahman Shiri, Barbara Bergbom

**Affiliations:** Finnish Institute of Occupational Health, P.O. Box 18, 00032 Helsinki, Finland

**Keywords:** absenteeism, disease management, health promotion, international cooperation, presenteeism, workplace

## Abstract

The aim of this scoping review was to identify effective workplace programs for work ability and well-being management and its barriers and facilitators in multinational organizations. The PubMed, Embase, and Web of Science databases were searched from 1974 through February 2023 to identify quantitative and qualitative studies on the management of work ability and well-being, and related outcomes including presenteeism, absenteeism, productivity loss, and healthy practices, conducted in a multinational organization or company. The titles and abstracts of over 11,000 publications were screened, and 10 studies fulfilling the inclusion criteria were included in the review. The management of work ability and well-being in multinational companies requires leadership support and commitment, effective communication, employee health awareness and engagement, comprehensive personalized health risk and condition assessments, and the management of risk factors and occupational and non-occupational health conditions. Financial constraints, high workloads, competing priorities, a lack of effective communication, a lack of worksite managers’ motivation, employees’ language barriers, high worksite managers’ turnover, and a decline in the support of senior managers are considered as barriers, and the presence of existing participatory practices is considered as a facilitator of participation in workplace health and well-being interventions in multinational companies. This review suggests that the management of work ability and well-being in multinational companies should go beyond health promotion and include comprehensive personalized health risk and health condition assessments and management.

## 1. Introduction

Globally, declines in work ability and productivity are growing concerns due to the ageing workforce. Work ability is a balance between work demands and individual resources consisting of training, competence, and functional ability [1]. Controlled clinical trials have found that workplace interventions such as exercise, mindfulness-based stress reduction programs, and ergonomics improvements prevent the decline in work ability and improve well-being [2,3,4,5,6,7,8].

Earlier studies have measured work ability using work performance, job effectiveness, presenteeism, absenteeism, or productivity loss [9,10,11]. Presenteeism, on-the-job productivity loss, is correlated with work ability and is the highest in workers with the lowest work ability [12]. Presenteeism is more costly than absenteeism (absence from work) [13]. The predictors of presenteeism and absenteeism include socio-demographic factors, lifestyle risk factors, and health problems [9,10,14]. Health problems, particularly depression and stress, are common causes of productivity loss [9]. Health promotion programs are implemented at workplaces to prevent presenteeism [14]. 

Various factors influence the implementation and delivery of workplace interventions for health promotion, well-being, and the prevention of a decline in work ability, which need to be considered when designing and implementing programs. Challenges and barriers to participation in workplace interventions, particularly among older workers, can reduce the efficacy of the intervention [15]. Knowledge of challenges, barriers, and facilitators can help organizations and employers to implement better health programs to improve work ability and well-being. Financial constraints, a lack of motivation, a lack of time, job stress, fatigue, and a lack of facilities at workplaces are reported as barriers to engagement in health promotion programs [16]. Adequate resources, the engagement of all stakeholders, effective leadership and guidance, trained staff, trust, and good communication are reported as facilitators of the implementation of health promotion programs [17].

Multinational companies adopt high-performance work practices more thoroughly than domestic companies [18]. However, the management of work ability and well-being is a challenging task in international companies due to the cultural diversity and differences in workplace health and safety legislation across countries [19]. There has been a significant growth in the number of multinational enterprises during the last century, and the number of parent multinational enterprises has increased from 7000 in 1970 to 38,000 in 2000 [20]. However, less is known about how multinational companies can best support their employees’ well-being and work ability. The purpose of this review was threefold: (1) to identify effective workplace programs for work ability and well-being management in multinational organizations, (2) to identify the challenges of work ability and well-being management in multinational organizations, and (3) to find common solutions discovered and implemented for the everyday management of work ability and well-being in multinational organizations.

## 2. Methods 

R.S. searched the PubMed, Embase, and Web of Science databases from 1974 through February 2023, using combinations of MeSH terms (PubMed), Emtree terms (Embase), and text words (Table 1). He also conducted an additional search in Google Scholar. There were no restrictions on the age or sex of participants, or the language of publications. Furthermore, the publications of the International Labour Organization (ILO) and the Organisation for Economic Co-operation and Development (OECD), and the reference lists of included articles on this topic, were also hand-searched for additional reports that might be relevant. R.S. screened the titles, abstracts, and full texts of relevant articles. A scoping review [21] was conducted and both quantitative and qualitative studies were included in the review. Eligible studies included randomized or non-randomized controlled trials; cross-sectional, case–control, and cohort studies; as well as qualitative studies that examined the management of work ability and well-being, and related outcomes including work performance, job effectiveness, productivity, presenteeism, absenteeism, and health promotion in a multinational organization or company. Furthermore, studies on challenges, barriers, and facilitators of work ability, well-being, or health promotion management in a multinational organization or company were also eligible. The results of the included quantitative studies were synthesized qualitatively. The methodological quality of the included studies was not evaluated in this review as the quality assessment was not relevant. 

## 3. Results 

R.S. screened 2941 publications in PubMed, 6150 in Embase, 2317 in Web of Science, and the first 1000 hits in Google Scholar. Ten studies (11 reports, two publications from one study) conducted in a multinational company were included in the review (Table 2). Some studies measured work ability using work performance [11], job effectiveness scores [10], or presenteeism [10]. Other studies examined safety, health, and well-being management [22,23,24]; changes in health risks [25]; the management of health risks and conditions [26]; absenteeism [10,11,27]; presenteeism [9], or the promotion of healthy practices [28,29].

Workers of multinational companies from developing countries more often reported more than one health risk factor for work ability and anxiety and/or depression than workers from developed countries [10]. On the other hand, workers from developed countries were on average older and less often satisfied with their jobs than workers from developing countries [10]. There were cross-country differences in work ability [10]. The work ability of employees of multinational companies was higher in Europe and North America, while it was lower in Asia and Latin America [10].

### 3.1. Management of Work Ability and Well-Being in International Organizations

A randomized controlled trial was conducted among older employees (aged 50 to 68 years) of a multinational information technology company to examine the efficacy of a web-based multimedia program in promoting healthy practices [28]. The intervention provided information, assessments, and recommendations regarding healthy aging, diet, smoking, physical activity, and stress management [28]. Three months following access to the program, the intervention had small beneficial effects on diet change self-efficacy, planning healthy eating, and mild exercise but had no effects on body mass index, eating practices, moderate or strenuous exercise, exercise planning, self-efficacy for overcoming barriers to exercise, symptoms of distress, coping with stress, and aging beliefs [28].

A health promotion and well-being program was developed to manage presenteeism at a multinational financial services company and consisted of raising health awareness and destigmatizing mental illness through multimedia learning, lunch and learn events, regular employee educational and focus group sessions, training supervisors and managers for health conditions, particularly for mental health, increasing employee participation in annual health risk assessments, improving employee engagement in healthy lifestyles (regular exercise, healthy nutrition, smoking cessation, avoiding excessive alcohol consumption, adequate sleep, and work–life balance), maintaining the company’s existing health practices and actions, and a supportive workplace health culture [9]. In a multinational food service company, a multi-level leadership team was involved in the implementation of an intervention for safety, health, and well-being to provide support and resources [22]. The following four strategies were recommended for the successful implementation of a safety, health, and well-being intervention: (1) leadership support and commitment to ensure resources and reinforce responsibility, (2) effective communication between organizational units, (3) engagement of stakeholders across all levels of the organization, and (4) customizing an intervention to a corporate environment [22]. 

A study conducted in a multinational agribusiness in Latin America examined the applicability of a health and well-being program using a total worker health approach within a multinational organization [24]. First, the culturally and linguistically appropriate worker health assessment was carried out at all levels of the organization and included (1) an organizational assessment in each country and each headquarters to evaluate the organization’s policies and activities for safety, health, and well-being; (2) a leadership assessment to evaluate leaders’ commitment to and promotion of worker health and well-being in their units; (3) an employee health and well-being culture assessment to examine employee’s perspectives regarding worker health at the organization using an online survey, and in-person interviews for those without access to a computer; (4) an employee health risk assessment using a questionnaire and employee clinic visit data, and (5) focus groups and interviews with key informants about the organization’s and leaders’ activities to promote and maintain worker health [24]. Second, necessary changes to the existing employee health and well-being policies and activities were recommended, and leadership training for company managers was developed and implemented at each national branch to help leaders to learn about and incorporate health and well-being actions into the company practices. This mixed methods study (quantitative and qualitative data) recommended the integration of health and well-being programs with workplace safety programs, and the prevention and management of sleep deprivation, fatigue, occupational stress, mental illness, and other occupational and non-occupational chronic diseases to improve employee health and well-being [24].

In a multinational chemical company, a combination of health awareness delivered through group sessions, mail, or during health assessments, and notifying employees about their health results and giving them advice to address their health problems, did not have significant beneficial effects on the health of the company’s employees in Latin America [25]. The company changed its health strategy and performed on-site comprehensive health assessments, provided vigorous personal risk assessments, and employee and healthcare staff together made specific plans for each risk factor and health condition, including treatment [25]. They also made follow-up plans and continued until the risk was controlled. With the new health strategy, the prevalence of various health risk factors such as systolic and diastolic blood pressure and total cholesterol markedly decreased, and the level of physical activity markedly increased [25]. Body mass index and the number of smokers also decreased [25]. The results indicate that the health strategies of multinational companies should not be limited to health promotion. They should include healthcare professionals with knowledge of the local culture [25]. Furthermore, another study recommended that the management of presenteeism and absenteeism should aim at reducing lifestyle risk factors; the management of pain, stress, sleep, and mental health; and improving job satisfaction [10].

In a quasi-experiment, a multicomponent workplace health promotion program improved work performance and productivity and reduced absenteeism among employees of a multinational manufacturing company [11]. The intervention also had beneficial effects on alcohol consumption, physical activity, nutrition, stress, sleep, and perceived general health [11]. The intervention consisted of a personalized health and well-being assessment; access to a health, well-being, and lifestyle improvement web portal; receiving an email every 2 weeks about relevant personal-well-being-related topics during a 12-month intervention period; and receiving a hard-copy newsletter and literature about stress management, sleep improvement, nutritional balance, and physical activity and on-site seminars about these topics [11]. The beneficial effects of the intervention on work performance did not differ by age, sex, salary, and the number of health risk factors at the baseline [11]. However, the intervention reduced absenteeism in women but not in men [11]. Moreover, the effect of the intervention on absenteeism was stronger among older women than among younger women [11]. 

### 3.2. Barriers and Facilitators

Occupational health services of multinational companies did not provide enough curative services for non-occupational diseases [26]. In a multinational manufacturing company, occupational health services personnel spent less than 10% of their time on curative services for non-occupational diseases in Belgium, France, Sweden, and the Netherlands, but spent more than 20% of their time in Germany, Hungry, Poland, Spain, Turkey, and UK [26]. Occupational health services personnel provided curative services for occupational diseases and injuries, periodic health check-ups for workers at high risk of harmful exposure, and health check-ups for workers returning from sick leave longer than three weeks and those who were involved in the surveillance of the work environment and control of hazards [26]. However, in some locations, they did not have a strategy for the systematic selection of workers at high risk of exposure for periodic health check-ups [26]. There were also variations in occupational health services within the same multinational company at different locations [26]. Occupational health services personnel were not sufficiently involved in developing workplace health strategies [26]. The involvement of occupational health services personnel in health promotion ranged between 1% and 20% in different locations [26].

In a multinational food service company, worksite managers were not adequately engaged with employees to implement safety, health, and well-being actions and the participation of employees in workplace interventions was also minimal [23]. High workloads and a lack of time were listed as barriers to participation in workplace health and well-being interventions for both worksite managers and employees [23]. Other barriers to participation in workplace health and well-being interventions included financial constraints, a fast-paced work environment, competing priorities, a lack of communication between worksite and senior managers, the lack of motivation of worksite managers, host company clients’ control over the worksite environment, employees’ language barriers, high worksite managers’ turnover, and a decline in senior managers’ support to worksite managers to overcome barriers to participation in workplace health and well-being interventions [22,23]. 

Existing participatory practices in a multinational company facilitated participation in workplace health and well-being interventions [23].

### 3.3. Solutions Discovered and Implemented

Knowledge of the available resources and benchmarks, and investment in social capital, should be considered in the development of a workplace health promotion program [9]. For the successful implementation of a health promotion and well-being program, mandatory training for supervisors and senior managers, particularly for mental health, is recommended [9]. A six one-day leadership training program to improve the psychological well-being of midlevel team leaders reduced both team leaders’ and team members’ absenteeism during and after the intervention in a health systems multinational organization [27]. Staff training and unit quality management systems were the most common organizational capacities for the implementation of a workplace health promotion program [29]. The following organizational capacities were recommended for the successful implementation of a health promotion program: (1) routine assessment of the quality of a health promotion program, (2) the presence of a fulltime health promotion coordinator, (3) the presence of an official health promotion team, and (4) officially documented strategies and standards for health promotion [29]. 

## 4. Discussion

This review suggests that the management of work ability and well-being in multinational companies requires employee health awareness and engagement, comprehensive personalized health risk and health condition assessments, and the management of health risks and conditions. Many companies do not provide health promotion and disease prevention services to their employees [30]. Workplace health promotion and disease prevention programs have beneficial effects on both employees and organizations in the long run [30,31,32]. They reduce employees’ health risks and conditions and the organization’s healthcare costs, absenteeism, and turnover rates [30,31,32]. The management of work ability and well-being in a large company requires the delivery of integrated occupational health services to address psychosocial and organizational risk factors and employees’ health risks and conditions [13].

The implementation of workplace work ability and well-being programs is challenging, particularly for older workers [15]. Older workers reported inadequate work environments and equipment, time pressure, high job demands, and little job rotation as barriers to the implementation of workplace work ability interventions [15]. Job demands and job strain were associated with lower work ability among employees of a multinational telecommunication company [33]. Work engagement decreased [34] and productivity loss increased [35] with an increase in the number of health risk factors, and workers who reduced their health risks, such as physical inactivity, excess body mass, smoking, excessive alcohol consumption, and stress, reduced their risks of presenteeism and absenteeism [35]. On the other hand, perceived supervisor support and good communication about internal organizational matters were associated with increased employee work engagement among workers of a multinational energy company [36]. Flexible work arrangements were associated with higher employees’ perceived productivity, and the associations were partly mediated by employee happiness [37]. Part-time work was also associated with higher productivity in older workers [38].

There are considerable cross-country differences in work performance and health risk factors and illnesses among employees of multinational companies [10]. The differences in workplace culture, such as practices and programs, leadership and management, support, and communication, need to be considered in the management of employee work ability and well-being in multinational companies. A prospective cohort study recruited patients with chronic back pain from Denmark, Germany, Israel, the Netherlands, Sweden, and USA and found large variations in the sustainable return-to-work after chronic back pain, ranging from 22% in Germany to 62% in the Netherlands [39]. Workplace interventions including the adaptation of the workplace and working hours, redesigning the job, and receiving benefits for the resumption of work were the most common reasons for between-nation differences. Moreover, job characteristics and eligibility criteria for disability benefits were other important contributors to between-nation differences [39]. Workplace ergonomics support improves workers’ safety and health [40], and workplace support for exercise, such as exercise programs, flexitime, showers, and bike storage, increases employees’ participation in the recommended levels of leisure time and commuting physical activity [41,42]. 

A high workload, time pressures, inflexibility of work, a lack of manager support, a lack of motivation, and not organizing health programs during work hours are also listed as barriers to participation in workplace health promotion and disease prevention programs in non-multinational enterprises [43,44]. Some barriers, such as a fast-paced work environment, competing priorities, host company clients’ control over the worksite environment, and employees’ language barriers, are more common in multinational enterprises than in national workplaces. 

This review indicates that work ability and well-being management, and its barriers and facilitators, have not been well studied in multinational companies. Only a limited number of studies on this topic were found. The management of work ability and well-being at workplaces requires the provision of indicators, tools, and actions. Culturally congruent strategies for the management of work ability and well-being in multinational companies should adopt comprehensive health risk assessments and routine health surveillance and management [30]. Adequate engagement of worksite managers in workplace health interventions is necessary, and their participation leads to their enhanced awareness of the importance of improvements in working conditions in employees’ safety, health, and well-being [23].

A strength of this review is the use of comprehensive, sensitive search terms and search strings in multiple large American and European databases. A large set of studies were screened to avoid missing relevant studies. However, the studies included in this review had some limitations. Most of the included studies were observational studies and only three studies used a randomized controlled trial or a quasi-experimental design. Some of the included studies developed a health and well-being program only for one national branch of a multinational company. The program may not be effective among the company’s employees in different countries. Furthermore, some of the included studies developed a health and well-being program for a multinational company, but the results of their program’s implementation were not available at the time of this review.

## 5. Conclusions

This scoping review indicates that the management of work ability and well-being in multinational companies should include health promotion, training supervisors and managers for health conditions, particularly mental health, and routine comprehensive personalized assessments and management of health risks and conditions. Moreover, it suggests that it is essential to ensure that cultural and linguistic barriers do not hinder the successful implementation of workplace interventions and programs to promote work ability and well-being in multinational companies. 

## Figures and Tables

**Table 1 healthcare-11-00978-t001:** PubMed, Embase, and Web of Science searches conducted on 8 February 2023.

Search	Query	No. of Items Found
**PubMed**		
#1	Workability [tiab] or “work ability” [tiab] OR “work disability” [tiab] OR health promotion [Mesh] OR wellbeing[tiab] OR “well-being” [tiab] OR absenteeism [Mesh] OR presenteeism[Mesh] OR productivity [tiab]	302,488
#2	“organization and administration” [Mesh] OR “organizations” [Mesh] OR “public sector” [Mesh] OR “private sector” [Mesh] OR enterprises [tiab] OR enterprise [tiab] OR company [tiab] OR companies [tiab] OR organisation * [tiab] OR organization * [tiab] OR corporate [tiab] OR corporation * [tiab] OR “human resource management” [tiab] OR “virtual teams”	2,482,299
#3	challenges [tiab] OR management [tiab] OR manager * [tiab] OR leadership [tiab] OR coordinat * [tiab] OR strateg * [tiab] OR process [tiab] OR processes [tiab] OR support service * [tiab] OR barriers [tiab] OR enablers [tiab] OR facilitators [tiab] OR supports [tiab] OR implement * [tiab]	5,775,485
#4	global [tiab] OR international [tiab] OR multinational [tiab] OR “multi-national” [tiab] OR “bi-national” [tiab] OR intercontinental [tiab] OR transnational [tiab]	911,085
#5	#1 AND #2 AND #3 AND #4	3603
Final	#5 Filters: Humans	2941

**Embase**		
#1	‘workability’ OR ‘work ability’ OR ‘work disability’/exp OR ‘health promotion’/exp OR ‘wellbeing’/exp/mj OR ‘absenteeism’/exp OR ‘presenteeism’/exp OR ‘productivity’/exp	206,221
#2	‘organization and management’/exp OR ‘organization’/exp OR ‘public sector’/exp OR ‘private sector’/exp OR ‘multinational corporation’/exp OR ‘enterprises’ OR ‘enterprise’ OR ‘company’ OR ‘companies’ OR ‘organisation *’ OR ‘organization *’ OR ‘corporate’ OR ‘corporation *’ OR ‘human resource management’ OR ‘virtual teams’	3,743,001
#3	‘challenges’ OR ‘management’ OR ‘manager *’ OR ‘leadership’ OR ‘coordinat *’ OR ‘strateg *’ OR ‘process’ OR ‘processes’ OR ‘support service *’ OR ‘barriers’ OR ‘enablers’ OR ‘facilitators’ OR ‘supports’ OR ‘implement *’	9,161,816
#4	‘global’ OR ‘transnational’ OR ‘multinational’ OR ‘multi-national’ OR ‘bi-national’ OR ‘intercontinental’ OR ‘international cooperation’/exp OR ‘international cooperation’	1,193,706
#5	#1 AND #2 AND #3 AND #4	6924
Final	#5 AND [humans]/lim	6150

**Web of Science**		
#1	TS = workability OR TI = workability OR AB = workability OR TS = “work ability” OR TI = “work ability” OR AB = “work ability” OR TS = “work disability” OR TI = “work disability” OR AB = “work disability” OR TS = ”health promotion” OR TI = ”health promotion” OR AB = ”health promotion” OR TS = wellbeing OR TI = wellbeing OR AB = wellbeing OR TS = “well-being” OR TI = “well-being” OR AB = “well-being” OR TS = absenteeism OR TI = absenteeism OR AB = absenteeism OR TS = presenteeism OR TI = presenteeism OR AB = presenteeism OR TS = productivity OR TI = productivity OR AB = productivity	570,187
#2	TS = “organization and administration” OR TI = “organization and administration” OR AB = “organization and administration” OR TS = “public sector” OR TI = “public sector” OR AB = “public sector” OR TS = “private sector” OR TI = “private sector” OR AB = “private sector” OR TS = enterprise OR TI = enterprise OR AB = enterprise OR TS = company OR TI = company OR AB = company OR TS = organisation * OR TI = organisation * OR AB = organisation * OR TS = corporate OR TI = corporate OR AB = corporate OR TS = corporation * OR TI = corporation * OR AB = corporation * OR TS = “human resource management” OR TI = “human resource management” OR AB = “human resource management” OR TS = “virtual teams” OR TI = “virtual teams” OR AB = “virtual teams”	665,671
#3	TS = challenges OR TI = challenges OR AB = challenges OR TS = management OR TI = management OR AB = management OR TS = manager * OR TI = manager * OR AB = manager * OR TS = leadership OR TI = leadership OR AB = leadership OR TS = coordinat * OR TI = coordinat * OR AB = coordinat * OR TS = strateg * OR TI = strateg * OR AB = strateg * OR TS = process OR TI = process OR AB = process OR TS = “support service *” OR TI = “support service *” OR AB = “support service *” OR TS = barrier OR TI = barrier OR AB = barrier OR TS = enabler OR TI = enabler OR AB = enabler OR TS = facilitator OR TI = facilitator OR AB = facilitator OR TS = support OR TI = support OR AB = support OR TS = implement * OR TI = implement * OR AB = implement *	15,207,767
#4	TS = global OR TI = global OR AB = global OR TS = multinational OR TI = multinational OR AB = multinational OR TS = “multi-national” OR TI = “multi-national” OR AB = “multi-national” OR TS = “bi-national” OR TI = “bi-national” OR AB = “bi-national” OR TS = intercontinental OR TI = intercontinental OR AB = intercontinental OR TS = transnational OR TI = transnational OR AB = transnational	1,385,207
Final	#1 AND #2 AND #3 AND #4	2317

**Table 2 healthcare-11-00978-t002:** Characteristics of the studies included in the review in chronological order.

Authors and Year of Publication	Study Population	Design	Sample Size	Intervention/Management	Outcome	Results
Management						
Jaramillo et al., 2021 [24]	A multinational agribusiness company	A mixed methods study	This study focused on company’s workers in three national branches in Guatemala, Nicaragua, and Mexico. Total of 1541 employees completed employee health and well-being culture assessment. Total of 142 department heads participated in focus groups and 29 corporate and department managers participated in interviews. Total of 167 leaders completed leadership self-sssessment and four lead human resource managers completed organizational assessment. Total of 120 leaders participated in the total worker health leadership training.	Total worker health approach consisting of (1) organization’s policies and activities in each country for employee safety, health, and well-being were assessed and necessary changes were recommended; (2) leaders’ commitment to and promotion of worker health and well-being were evaluated and leadership training for managers was developed and implemented at each national branch; (3) employee health and well-being culture and employee health risk were assessed, and (4) clinical consult data were evaluated for diagnoses and reasons for medical visits.	The development of an organizational program and policy for employee health and well-being	The study recommended the integration of health and well-being programs with workplace safety programs, prevention and management of occupational and non-occupational chronic diseases, sleep deprivation, fatigue, occupational stress, and mental illness.
Van Tuin et al., 2020 [27]	A health systems multinational organization	A quasi-experiment A 12-month follow-up	N = 119 (13 team leaders and 106 team members) for intervention group N = 158 (39 team leaders and 119 team members) for control group	A leadership training program to improve psychological well-being of midlevel team leaders	Sickness absenteeism	The leadership training program reduced both team leaders’ and team members’ absenteeism over 12 months after the intervention.
Howarth et al., 2017 [10]	254 multi-national companies from 120 countries	Cross-sectional	117,274 (N = 87,170 from 32 developed countries and N = 30,104 from 88 developing countries)	Health risk assessments	Job effectiveness score, absenteeism, and presenteeism	There were cross-country differences in job effectiveness, and health risk factors and illnesses. Region of residency, stress, perceived general health, job satisfaction, and pain were the most important determinants of presenteeism, and pain, age, perceived general health, and stress were most important determinants of absenteeism.
Ammendolia et al., 2016 [9]	A multinational financial services company	An intervention mapping. Administrative and claims data, summary results of company’s online employee wellness surveys (2009, 2010), plus 4 interviews and 4 focus group discussions were utilized.	Overall, 31% of over 8000 employees of company participated in 2019 online survey and 12% in 2010 survey. Total of 37 participants took part in focus group discussions.	To develop a health promotion and well-being program to manage presenteeism	Presenteeism	The program consisted of raising health awareness and destigmatizing mental illness, training supervisors and managers on health conditions, particularly for mental health, increasing employee participation in annual health risk assessment, improving employee engagement in healthy lifestyles, and a supportive workplace health culture.
Cook et al., 2015 [28]	A multinational information technology company	A randomized controlled trial A 3-month follow-up	278 (138 intervention group and 140 control group)	A web-based multimedia health promotion program consisting of healthy aging, diet, smoking, physical activity, and stress management	Promoting healthy practices	The program had small beneficial effects on diet change self-efficacy, planning healthy eating, and mild exercise but had no effects on body mass index, eating practices, moderate or strenuous exercise, exercise planning, self-efficacy for overcoming barriers to exercise, symptoms of distress, coping with stress, and aging beliefs.
Rager et al., 2008 [25]	A multinational chemical company	Repeated cross-sectional study	Not reported	Employee disease management consisting of onsite comprehensive health assessments, and management of health risks and conditions	Changes in health risks	Only health awareness and notification of employee health results and provision of advice to address the health problem had no significant beneficial effects on employee health. With both the assessment and management of health risks and conditions, the prevalence of systolic and diastolic blood pressure, total cholesterol, and physical inactivity decreased by at least 58% and smoking and body mass index decreased by over 10%. The study recommended that multinational companies’ health strategies should not be limited to health promotion. They should also include healthcare professionals with knowledge of local culture.
Mills et al., 2007 [11]	A multinational manufacturer of food, home care, and personal care products	A quasi-experiment A 12-month follow-up	N = 266 for intervention group and N = 1242 for control group	A multicomponent workplace health promotion program consisting of a personalized health and well-being assessment; access to a tailored health, well-being, and lifestyle improvement web portal; literature, and seminars and workshops	Number of health risk factors, work performance, and absenteeism	The number of health risk factors and absenteeism days were significantly lower, and work performance was significantly higher in intervention than control group.
Bratveit et al., 2001 [26]	A multinational manufacturing company	A qualitative study	Questionnaire for 20 occupational health services units, structured interviews, and site visits to 20 locations in 11 countries	Activities of the occupational health services	Management of employee health risks and conditions	Occupational health services personnel of multinational companies provided curative services for occupational diseases and injuries, periodic health check-ups for workers at high risk of harmful exposure, and health check-ups for workers returning from sick leave longer than three weeks and those who were involved in surveillance of work environment and control of hazards. However, they did not provide enough curative services for non-occupational diseases, and in some locations, they did not have a strategy for the systematic selection of workers at high risk of harmful exposure for periodic health check-ups. Moreover, they were not sufficiently involved in workplace health promotion programs.

Barriers and facilitators						
Roodbari et al., 2022 [23] and Sorensen et al., 2021 [22]	A multinational food service company	A cluster randomized controlled trial	A qualitative analysis of 25 interviews and 89 process tracking documents from the intervention sites	A multi-level leadership team to provide support and resources consisting of (1) leadership support and commitment, (2) effective communication between units, (3) engagement of stakeholders across all levels of the organization, and (4) customizing an intervention to corporate environment	Facilitators and barriers to participation in safety, health, and well-being intervention	Existing participatory practices facilitated participation in workplace interventions [23]. Barriers to participation in workplace intervention included financial constraints, the lack of motivation of worksite managers, employees’ language barriers, high worksite managers’ turnover, fast-paced work environment, competing priorities, lack of communication between worksite and senior managers, host company clients’ control over the worksite environment, and decline in senior managers’ support to worksite managers to overcome barriers [22,23].
Solutions discovered for successful implementation						
Röthlin et al., 2013 [29]	The International Network of Health Promoting Hospitals and Health Services	Cross-sectional study	29 national and regional networks (159 hospitals)	Capacities for the implementation of health promotion	Promoting healthy practices	The following organizational capacities were recommended for successful implementation of a health promotion program: (1) routine assessment of the quality of a health promotion program, (2) presence of a fulltime health promotion coordinator, (3) presence of an official health promotion team, and (4) officially documented strategies and standards for health promotion.

## Data Availability

Not applicable.

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
