# Peer review of "Work Ability and Well-Being Management and Its Barriers and Facilitators in Multinational Organizations: A Scoping Review"

_healthcare, 2023, doi:10.3390/healthcare11070978_

Round 1

Reviewer 1 Report

I would like to congratulate the authors for their interest in researching in this field, however, the work presented presents some deficiencies.

1. Authors should take care in the presentation of the document. There are several sections with different font sizes (example line 54, 233, 242,…). The authors should review the document and correct these errors.

2. Authors should avoid the use of the first grammatical person, both singular and plural, in the writing of research articles (example, line 259). They should replace this verb tense with impersonal tenses. The direct references to one of the authors regarding the work they have developed in this review are not correct either (example line 68, 75, 89,…).

3. Introduction chapter is correct, although I believe that the references used could be expanded in order to provide the reader with the context of this review.

4. The authors perform a quantitative analysis of the references to the subject matter of this review. However, in my opinion, this analysis is not sufficient to justify the present article.

The rest of the aspects described in chapter 3, although interesting and correctly described, should be expanded, given that this is the main section that contributes value to the reader.

5. Conclusions chapter should be revised. On the other hand, the authors have included numerous conclusions in the discussion section.

Conclusions chapter should allow the reader to know in a synthetic way, the main contributions of this review so that readers can obtain this information in a summarized form.

I hope that these changes will help to improve your article and make it a document of great scientific interest.

Author Response

Reviewer #1

I would like to congratulate the authors for their interest in researching in this field, however, the work presented presents some deficiencies.

Response. Thank you for your comments!

  1. Authors should take care in the presentation of the document. There are several sections with different font sizes (example line 54, 233, 242,…). The authors should review the document and correct these errors.

Response. We have corrected font sizes and errors. The publisher takes care of those issues before publishing the report.

  1. Authors should avoid the use of the first grammatical person, both singular and plural, in the writing of research articles (example, line 259). They should replace this verb tense with impersonal tenses. The direct references to one of the authors regarding the work they have developed in this review are not correct either (example line 68, 75, 89,…).

Response. Some journals prefer active sentences, and some journals prefer passive sentences. We used active sentences to indicate that only the first author screened the titles and abstracts but not both authors. Moreover, it indicates that the first author takes responsibility for accuracy of screening and data extraction.

  1. Introduction chapter is correct, although I believe that the references used could be expanded in order to provide the reader with the context of this review.

Response. This topic is not well studied. We cited most of the relevant studies (20 references) in the introduction.

  1. The authors perform a quantitative analysis of the references to the subject matter of this review. However, in my opinion, this analysis is not sufficient to justify the present article.

The rest of the aspects described in chapter 3, although interesting and correctly described, should be expanded, given that this is the main section that contributes value to the reader.

Response. We expanded the results and discussion sections and included a table in the manuscript to describe the characteristics and results of the included studies.

  1. Conclusions chapter should be revised. On the other hand, the authors have included numerous conclusions in the discussion section.

Conclusions chapter should allow the reader to know in a synthetic way, the main contributions of this review so that readers can obtain this information in a summarized form.

Response. We have revised the conclusions on pages 1 and 16.

I hope that these changes will help to improve your article and make it a document of great scientific interest.

Reviewer 2 Report

This study presents a review of the literature relevant to the topic, however taking into account the diversity and plurality of aspects under study, it should segment by aspects or factors.

Improvement proposals

Line 9 – „were searched from their inception through September 2022. Authors must describe their inception.

Line 69 – “R.S. searched PubMed, Embase and Web of Science databases from their inception through September 2022” The authors must justify the choice of these databases for the purpose of the study.

Line 246 - The cultural factors need to be considered in the management of work ability and well-being. Authors should describe the cultural factors and justify their choice.

Line 259 – “We found only a limited number of studies on this topic. How many should authors specify?

The authors must conclude, in a descriptive way, the main considerations, ideas, studies and aspects observed during the study. They must also respond objectively to the purposes of this study -

“The purpose of this review was threefold: 1) to identify effective workplace programs for work ability and well-being management in multinational organizations, 2) to identify the challenges of work ability and well-being management in multinational organizations, and 3) to find out common solutions discovered and implemented for everyday management of work ability and well-being in multinational organizations.”

The limitations of this study and future studies must also be mentioned.

Author Response

Reviewer #2

This study presents a review of the literature relevant to the topic, however taking into account the diversity and plurality of aspects under study, it should segment by aspects or factors.

Response. Thank you for your comments!

Improvement proposals

Line 9 – „were searched from their inception through September 2022. Authors must describe their inception.

Response. Inception is the starting year of database and is different for each database. We have change “inception” to “1974”. We did not find any publication on this topic before 1974.

Line 69 – “R.S. searched PubMed, Embase and Web of Science databases from their inception through September 2022” The authors must justify the choice of these databases for the purpose of the study.

Response. A comprehensive literature search should include at least one American and one European database. Web of Science is the largest American database and PubMed is the second largest American database. Embase is one of the two largest European databases.

Line 246 - The cultural factors need to be considered in the management of work ability and well-being. Authors should describe the cultural factors and justify their choice.

Response. We have described them on page 15.

Line 259 – “We found only a limited number of studies on this topic. How many should authors specify?

Response. In addition to work ability and well-being, we included other outcomes such as presenteeism and absenteeism. We found only 10 studies. Ten studies are not even enough for a single outcome.

The authors must conclude, in a descriptive way, the main considerations, ideas, studies and aspects observed during the study. They must also respond objectively to the purposes of this study -

“The purpose of this review was threefold: 1) to identify effective workplace programs for work ability and well-being management in multinational organizations, 2) to identify the challenges of work ability and well-being management in multinational organizations, and 3) to find out common solutions discovered and implemented for everyday management of work ability and well-being in multinational organizations.”

Response. We have described the results in three subsections to answer the three review questions. However, we found only one study (two reports) to examine the second review question and one study to examine the third review question. The aim of a scoping review is to identify the nature and extent of research evidence and gaps in existing literature. We have reported the volume and nature of studies conducted in multinational companies to manage work ability and well-being and described the characteristics and finding of the included studies.

The limitations of this study and future studies must also be mentioned.

Response. We have discussed the limitations of the studies included in the review on page 16.

Reviewer 3 Report

The article presents a relevant goal. The article presents a scoping review that should be methodologically grounded and supported in a more structured research protocol.  The transition from 9,000 publications to 10 studies is not methodologically explicit.The article does not explicitly adopt a systematic methodology in the preliminary post-bibliographical database search phase. The contributions that can be drawn from the results of the research carried out are incipient and lack a more in-depth (content analysis) and a more sophisticated data treatment and analysis in terms of the form of presentation and scientific dissemination.

The significant contribution to the understanding of barriers and facilitators is not evident, nor is the contribution in the context of multinational companies. The article's contributions to the subject area are not significant. Significant further development and deepening is required.

Author Response

Response. Thank you for your comments!

This is a scoping review and is different from narrative, systematic, umbrella, and mixed methods reviews. A scoping review identifies nature and extent of research evidence and gaps in existing literature. We used sensitive search terms and sensitive search strings and screened a large number of studies to limit missing any eligible study. Of 11,000 publications on this topic, only 10 studies were conducted in a multinational company. This is a strength of this review that we screened 11,000 publications to avoid missing a relevant study. This review reports the volume and nature of studies conducted in multinational companies to manage work ability and well-being and describes the characteristics and finding of the included studies. We expanded the results and discussion sections and included a table in the manuscript to describe the characteristics and results of the included studies.

Round 2

Reviewer 2 Report

The article is better now

Author Response

Thanks for your positive comments!